# eQTL Catalogue 2023: New datasets, X chromosome QTLs, and improved detection and visualisation of transcript-level QTLs

Nurlan Kerimov[1,2], Ralf Tambets[1], James D. Hayhurst[2,3], Ida Rahu[1], Peep Kolberg[1], Uku Raudvere[1], Ivan Kuzmin[1], Anshika Chowdhary[4], Andreas Vija[1], Hans J. Teras[1], Masahiro Kanai[5,6,7], Jacob Ulirsch[5,6,7], Mina Ryten[8], John Hardy[8], Sebastian Guelfi[8], Daniah Trabzuni[8], Sarah Kim-Hellmuth[4,9], William Rayner[4], Hilary Finucane[5,6,7], Hedi Peterson[1], Abayomi Mosaku[2,3], Helen Parkinson[2,3], Kaur Alasoo[1,2]*

**1** Institute of Computer Science, University of Tartu, Tartu, Estonia, **2** Open Targets, South Building, Wellcome Genome Campus, Hinxton, Cambridge, United Kingdom, **3** European Molecular Biology Laboratory, European Bioinformatics Institute, Wellcome Genome Campus, Hinxton, Cambridge, United Kingdom, **4** Institute of Translational Genomics, Helmholtz Munich, Neuherberg, Germany, **5** Analytic and Translational Genetics Unit, Massachusetts General Hospital, Boston, Massachusetts, United States of America, **6** Program in Medical and Population Genetics, Broad Institute of MIT and Harvard, Cambridge, Massachusetts, United States of America, **7** Stanley Center for Psychiatric Research, Broad Institute of MIT and Harvard, Cambridge, Massachusetts, United States of America, **8** Department of Genetics and Genomic Medicine, Great Ormond Street Institute of Child Health, University College London, London, United Kingdom, **9** Department of Pediatrics, Dr. von Hauner Children's Hospital, University Hospital LMU Munich, Munich, Germany

* kaur.alasoo@ut.ee

**Data Availability Statement:** The molecular QTL summary statistics, fine mapping results (including SuSiE log Bayes factor) and QTL coverage plots are

## Abstract

The eQTL Catalogue is an open database of uniformly processed human molecular quantitative trait loci (QTLs). We are continuously updating the resource to further increase its utility for interpreting genetic associations with complex traits. Over the past two years, we have increased the number of uniformly processed studies from 21 to 31 and added X chromosome QTLs for 19 compatible studies. We have also implemented Leafcutter to directly identify splice-junction usage QTLs in all RNA sequencing datasets. Finally, to improve the interpretability of transcript-level QTLs, we have developed static QTL coverage plots that visualise the association between the genotype and average RNA sequencing read coverage in the region for all 1.7 million fine mapped associations. To illustrate the utility of these updates to the eQTL Catalogue, we performed colocalisation analysis between vitamin D levels in the UK Biobank and all molecular QTLs in the eQTL Catalogue. Although most GWAS loci colocalised both with eQTLs and transcript-level QTLs, we found that visual inspection could sometimes be used to distinguish primary splicing QTLs from those that appear to be secondary consequences of large-effect gene expression QTLs. While these visually confirmed primary splicing QTLs explain just 6/53 of the colocalising signals, they are significantly less pleiotropic than eQTLs and identify a prioritised causal gene in 4/6 cases.

available from the eQTL Catalogue FTP server (see https://www.ebi.ac.uk/eqtl/Data_access/). The marginal eQTL summary statistics are also available via our REST API (https://www.ebi.ac.uk/eqtl/api/docs), which we have completely re-written for release 6. RNA-seq and genotype data from the CAP (phs000481.v3.p2), Peng_2018 (phs001586.v1.p1), PhLiPS (phs001341.v1.p1) and iPSCORE (phs000924.v4.p1) studies were downloaded from dbGaP; Steinberg_2020 (EGAD00001005215, EGAD00001003355, EGAD00010001746), Young_2019 (EGAD00001005736), Braineac2 (EGAD00001005526, EGAD00001003100) and Bossini-Castillo_2019 (EGAD00001004830, EGAD00010001848) from EGA, and CommonMind (syn2759792) from Synapse. Raw genotype data for Gilchrist_2021 (EGAD00010000144, EGAD00010000520) was downloaded from EGA. Raw gene expression data from Gilchrist_2021 was downloaded from Zenodo (https://doi.org/10.5281/zenodo.6352656).

**Funding:** N.K., J.D.H., P.K. and H.J.T. were supported by a grant from Open Targets (grant no. OTAR2077). H. P., A.M. and J.H. were supported by the European Molecular Biology Laboratory. K.A., A.C., W.R. and N.K. also received funding from the European Union's Horizon 2020 research and innovation program (grant no. 825775). K.A., N.K, R.T., A.V., P.K. and I.R. were supported by the Estonian Research Council (grant no. PSG415). K.A. was supported by the Estonian Research Council (grant no. IUT34-4). K.A. and N.K. were also supported by the Estonian Centre of Excellence in ICT Research (EXCITE), funded by the European Regional Development Fund. I.K., U.R. and H. P. were supported by the Distributed Infrastructure for Life-Science Information ELIXIR, European Regional Development Fund project (2014-2020.4.01.16-0271). S.K.-H. was supported by the Emmy Noether Programme KI 2091/2-1 (459153572), SFB/TRR237-B29 (369799452) and SFB/TRR359-B06 (491676693) of the Deutsche Forschungsgemeinschaft (DFG). Funding information for individual studies included in the eQTL Catalogue is presented in S1 Text. The funders had no role in study design, data collection and analysis, decision to publish, or preparation of the manuscript.

**Competing interests:** I have read the journal's policy and the authors of this manuscript have the following competing interests: J.C.U. is an employee of Illumina. N.K. is an employee of Nightingale Health. S.G. is an employee of Verge Genomics.

## Author summary

Genome-wide association studies have identified many non-coding loci associated with complex traits and diseases. While these variants are primarily expected to affect gene regulation, identifying the target genes as well as the tissue and cell type contexts where these variants are active has remained challenging. We have previously developed the eQTL Catalogue resource to systematically curate and reprocess all available RNA sequencing and genotype datasets. Here we present several updates to the resources that increase its utility for interpreting complex trait associations even further. In addition to increasing both the number of studies and datasets covered, we have also implemented statistical fine mapping to improve our ability to link multiple independent genetic variants influencing gene regulation with complex traits and diseases. We demonstrate the utility of these updates by focussing on interpreting genetic variants associated with a single well-characterised complex trait—circulating levels of vitamin D in human plasma. We believe that our publicly available analysis results will greatly facilitate the interpretation of complex trait associations identified by other large-scale human genetics efforts.

## Introduction

Most genetic variants associated with complex traits are in the non-coding regions of the genome [1]. More than a decade of molecular quantitative trait locus (QTL) studies have revealed that these variants regulate either the expression level [2,3], splicing [4], promoter usage [5,6] or alternative polyadenylation [7,8] of their target genes. Although the eQTL Catalogue has contained transcript-level QTL summary statistics from the beginning, characterising the exact mechanism of action of each molecular QTL has remained challenging due to considerable overlap between QTLs detected by different RNA-seq quantification methods [2], technical biases in read alignment [9], and a large number of alternative transcripts or splice junctions to be considered for each gene. Furthermore, because the usage of each transcript or splice junction is quantified relative to all other transcripts, the magnitude and direction of the genetic effect, the part of the gene affected, as well as the absolute expression of the affected transcript is often difficult to assess from summary statistics alone.

This ambiguity can be reduced by visualising the change in the average RNA-seq read coverage in the gene region associated with each additional copy of the alternative allele. We and others have used these QTL coverage plots to characterise chromatin QTLs [10–12] as well as to confirm promoter usage and splicing QTLs [5]. However, previous studies have visualised individual molecular QTLs in a setting where access to individual-level genotype and read coverage data is available. It has not been done systematically in large molecular QTL compendia such as the GTEx project [3] and the eQTL Catalogue, because in a naive implementation the read coverage stratification by genotype needs to be performed separately for each significant genetic variant and molecular trait pair of interest. Since transcript and exon-level analyses profile hundreds of thousands of correlated molecular traits in a single dataset, this means that the number of QTL coverage plots required can quickly become intractable.

In this update to the eQTL Catalogue, we present an approach to generate QTL coverage plots for all independent genetic signals and their associated molecular traits. First, we have updated our data processing workflows to improve promoter usage and splicing QTL discovery and to generate read coverage signals for all 25,724 RNA-seq samples. We have also adopted fine-mapping-based filtering to identify all independent genetic signals and associated molecular traits for each gene while reducing the size of the summary statistics files by 98%. Finally, to support new colocalisation methods that can account for multiple independent

causal variants [13], we have computed signal-level log Bayes factors for all independent signals [14]. This approach has enabled us to predefine tag variants and molecular traits for all independent genetic associations identified in 127 eQTL datasets and generate QTL coverage plots that can be used to interpret almost all colocalising signals detected in the eQTL Catalogue.

## Results

### Updates to the eQTL Catalogue resource

The aim of the eQTL Catalogue is to provide a public resource of uniformly processed molecular QTL summary statistics and continuously update this resource as new studies, reference annotations and quantification methods become available. Here, we present the updates to the eQTL Catalogue release 6 that we have made since the publication of the original paper (release 3).

**Newly added datasets.** We have added nine new RNA-seq studies and one microarray study to the eQTL Catalogue. This has increased the total number of studies in the resource to 31, the total number of datasets to 127 and the cumulative number of donors and samples to 7,526 and 30,602, respectively (Fig 1A). Newly added datasets include additional datasets from tissues and cell types already present in the eQTL Catalogue (e.g. various brain regions [15,16], immune cells [17–20] and induced pluripotent stem cells [21,22]) as well as previously missing microglia [23], placenta [24], hepatocytes [22], and cartilage and synovium tissues [25]. Complete summary of the datasets present in the eQTL Catalogue is shown in S1 Table.

**Imputation of the X chromosome genotypes.** In addition to integrating new datasets, we also made two major changes to our genotype imputation workflow. First, we migrated to the

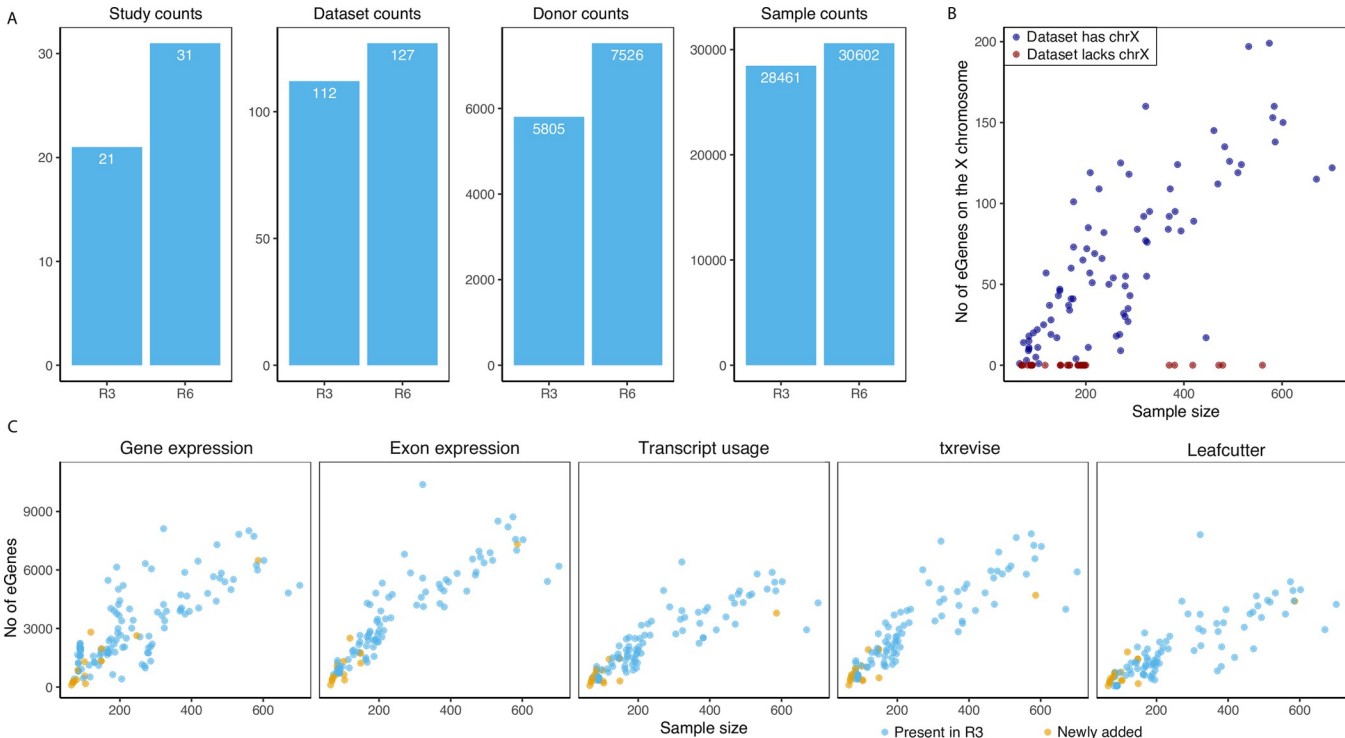

**Fig 1. Uniform re-processing of all datasets.** (**A**), The number of studies, datasets, donors and samples in the previous publication (R3) and current version of the eQTL Catalogue (R6). (**B**), Number of genes with at least one significant eQTL ('eGenes') on the X chromosome as a function of dataset sample size. Red points indicate datasets for which the X chromosome genotypes were unavailable. (**C**), The number of eGenes identified in each dataset for the five molecular traits (gene expression, exon expression, transcript usage, txrevise event usage, and Leafcutter splice-junction usage). Datasets newly added since release 3 have been highlighted.

new 1000 Genomes 30x of GRCh38 reference panel [26]. This allowed us to impute genotypes directly to the GRCh38 build and avoid errors caused by the genomic coordinate lift over process. Secondly, our imputation workflow now also supports the X chromosome. As a result, summary statistics for 18 of the 31 studies now also contain variants from the X chromosome. Across these 18 datasets, we detected at least one significant eQTL (FDR <1%) for 853 unique genes on the X chromosome (Fig 1B). These X chromosome eQTLs account for ~1.6% of all significant eQTLs (FDR < = 1%). Ten of the other 13 studies are missing the X chromosome QTLs because X chromosome genotypes were not deposited with data. Exceptions are male only studies (n = 3) that did not pass our genotype QC criteria (S2 Table).

**Improved quantification of splicing and promoter usage QTLs.** The previous release of the eQTL Catalogue included four molecular trait quantification methods to measure transcriptional changes from RNA-seq data: gene expression (ge), exon expression (exon), transcript usage (tx) and transcriptional event usage (txrevise). In addition to these four, we have now also implemented LeafCutter [27] to directly quantify the usage of splice junctions (Fig A in S1 Text). We have also augmented the txrevise promoter annotations with experimentally determined promoters from the FANTOM5 project [28]. Finally, we have updated the reference transcriptome annotations to Ensembl version 105 and GENCODE version 39. We observed a clear linear relationship between the number of significant associations detected with each quantification method and the dataset sample size, with gene expression, exon expression and txrevise detecting, on average, slightly more associations than transcript usage and Leafcutter (Fig 1C). For gene expression QTLs, we further quantified the number of novel associations detected in release 6 relative to release 3. The number of unique genes with at least one eQTL ('eGenes') increased by 1.8% whereas the number of confidently fine mapped eQTL gene-variant pairs (posterior inclusion probability > 0.95) increased by 4% (Fig B in S1 Text). Most of this increase was driven by the CommonMind [16] prefrontal cortex dataset that had the largest sample size (n = 586).

**Fine-mapping-based filtering of transcript-level summary statistics.** A major challenge in working with exon- and transcript-level (transcript usage, txrevise, leafcutter) associations is the large number of correlated traits being tested that result in very large summary statistics files. For example, typical summary statistics for exon and txrevise QTLs are 15–20 times larger than the corresponding files for gene expression QTLs. In addition to complicating our data release and archival procedures, these large file sizes meant that performing comprehensive colocalisation analysis against the eQTL Catalogue required the downloading and processing of >15Tb of data. To reduce the size of these files, we have now implemented fine-mapping based filtering. Briefly, we are using fine mapped credible sets to identify all independent signals at the gene level. We then filter the summary statistics files to only retain the most strongly associated molecular trait (exon, transcript, txrevise event or Leafcutter splice junction) for each signal. This filtering reduces the size of the summary statistics files for those quantification methods by ~98% while retaining the vast majority of significant associations for colocalisation purposes. Reducing the size of the univariate summary statistics files has also allowed us to export SuSiE log Bayes factors for each fine mapped signal and all tested variants [14]. As illustrated below, these log Bayes factors can be directly used in the new coloc.susie method to perform colocalisation analysis between all pairs of independent signals [13].

**Visualisation of transcript-level associations.** Another benefit of fine-mapping-based filtering is that we now have a tractable set of independent lead variants and associated molecular traits across all datasets and quantification methods that we can visualise using static QTL coverage plots. These plots display normalised RNA-seq read coverage across all exons of the gene (Fig 2A), exon-level QTL effect sizes and standard errors (Fig 2B), as well as the alternative transcripts or splice junctions used in association testing (Fig 2C). As an example, we are

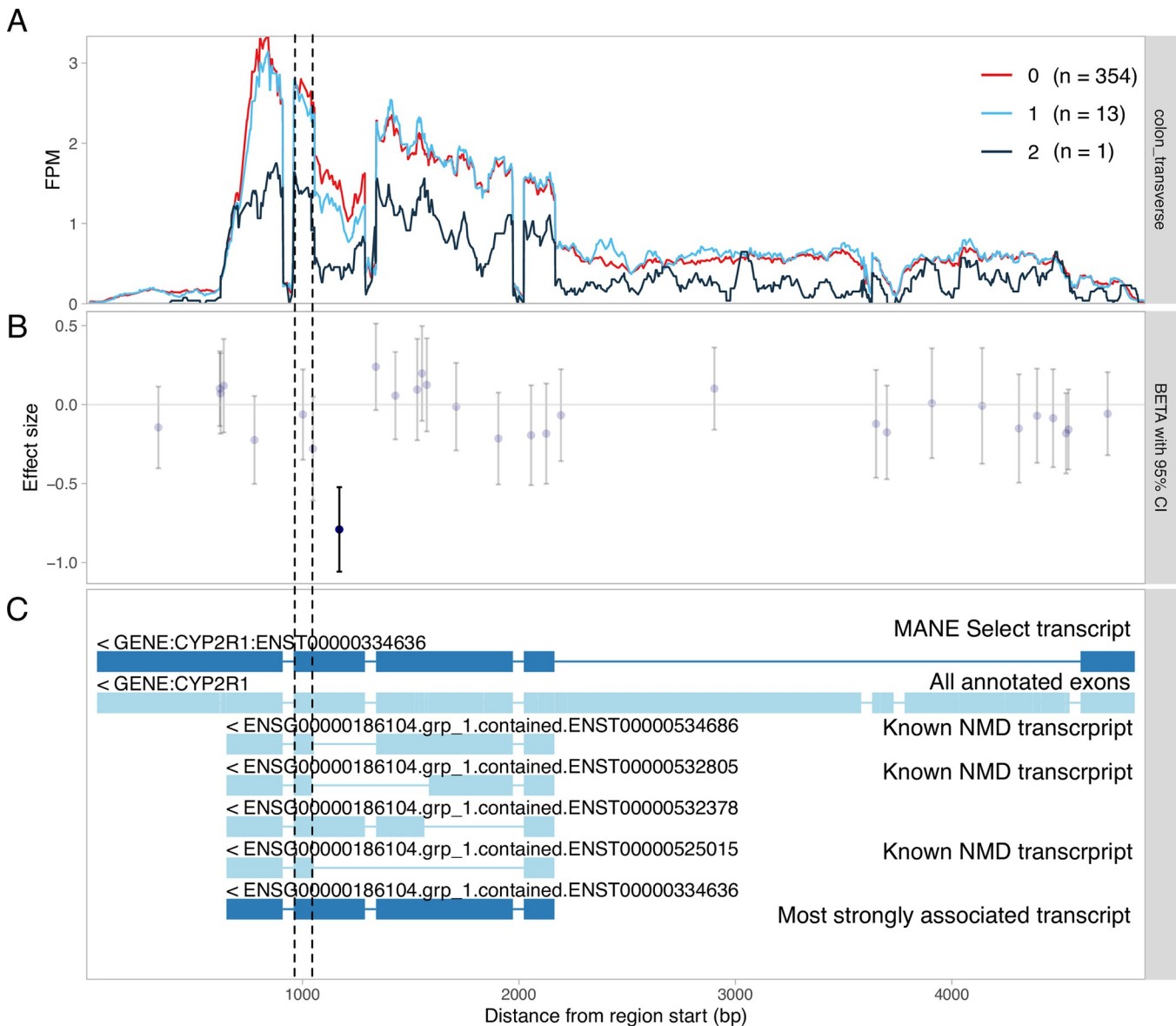

**Fig 2. Visualisation of a splicing QTL detected in the *CYP2R1* gene.** (**A**) RNA-seq read coverage across the *CYP2R1* gene in GTEx transverse colon tissue stratified by the genotype of the lead sQTL variant (chr11_14855172_G_A). All introns have been shortened to 50 nt with wiggleplotr [29] to make variation in exonic read coverage easier to see. (**B**) Effect sizes and 95% confidence intervals of the lead sQTL variant on the expression level of individual exons (or exonic parts) of *CYP2R1*. Associations significant at FDR < = 1% are shown in dark blue. (**C**) The top two rows show the MANE Select [30] reference transcript and all annotated exons of *CYP2R1*, respectively. The remaining rows show the txrevise [5] event annotations used for sQTL mapping. The short version of exon 4 (between dashed lines) is only present in annotated nonsense-mediated decay (NMD) transcripts.

highlighting the association between chr11_14855172_G_A and alternative splicing of exon four of the *CYP2R1* gene (Fig 2). The static QTL coverage plots for all 1,716,482 independent signals are now available via the eQTL Catalogue FTP server.

## Case study: Target gene prioritisation for vitamin D GWAS

To test the utility of the new QTL coverage plots, we performed a proof-of-concept colocalisation analysis between all molecular traits in the eQTL Catalogue and vitamin D levels in the

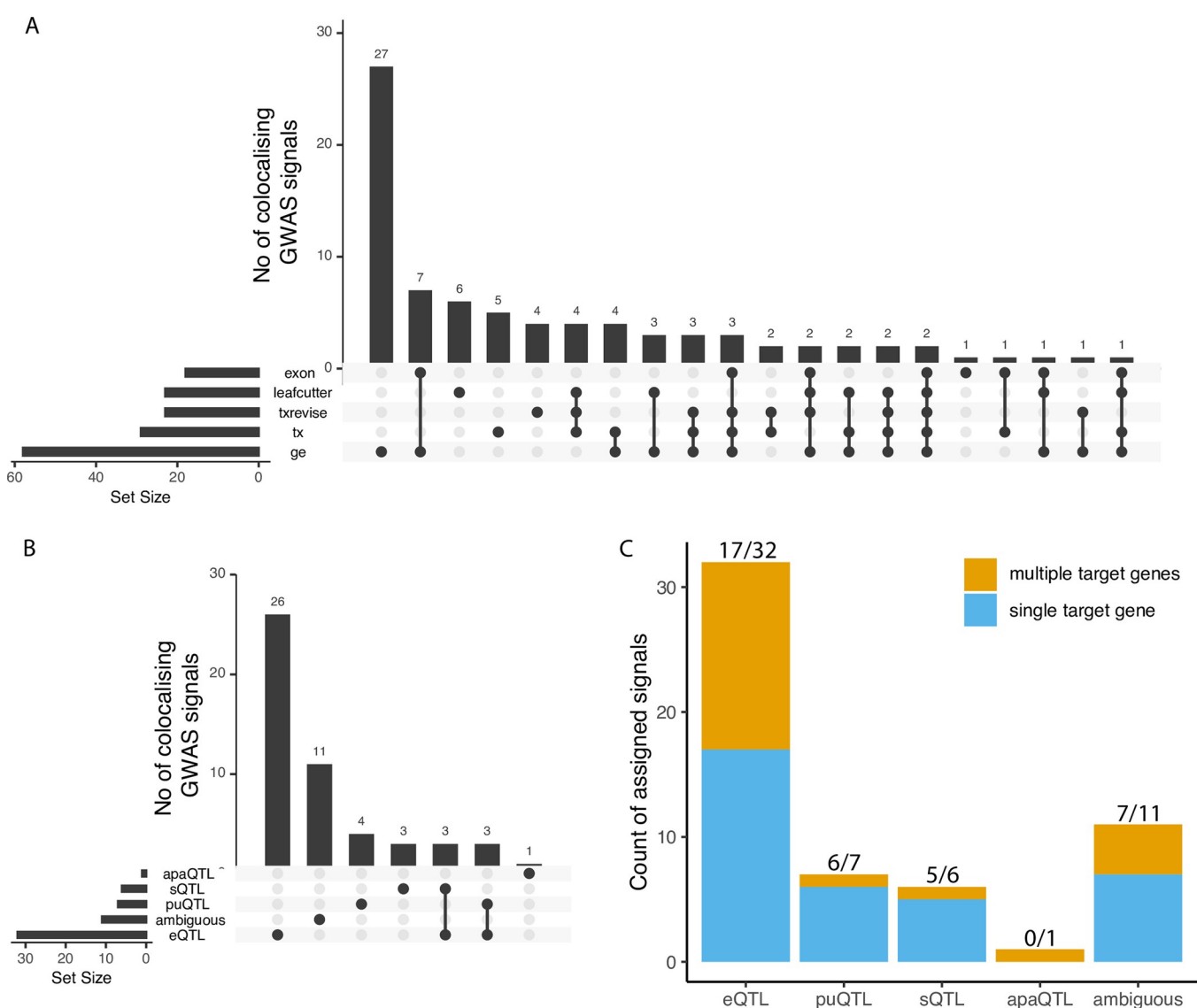

**Fig 3. Sharing of significantly colocalised signals with vItamin D.** (**A**) Number of colocalised signals detected by the different molecular QTL quantification methods and sharing between them. (**B**) Number of colocalised signals assigned to empirical functional consequence (eQTL, sQTL, puQTL, apaQTL or ambiguous) and sharing structure between them. (**C**) Number of independent colocalised signals associated with either a single target gene or multiple target genes in each functional consequences group. eQTL—expression QTL, sQTL—splicing QTL, puQTL—promoter usage QTL, apaQTL—alternative polyadenylation QTL.

UK Biobank. We chose this phenotype, because the vitamin D biosynthesis pathway is well understood and many causal genes underlying GWAS associations for vitamin D are already known [31,32]. At a stringent threshold of PP4 > 0.9, we found that 53/83 signals from 34/48 regions colocalised with 81 protein coding genes (Fig 3A). Although colocalisation with total gene expression was most commonly observed, there was considerable overlap between colocalisations detected with the five different quantification methods (Fig 3A).

We extracted QTL coverage plots for all 816 colocalising molecular QTL signals. We then manually reviewed the plots to classify each signal into one of five categories: expression QTLs, promoter usage QTLs (puQTLs), splicing QTLs (sQTLs), alternative polyadenylation QTLs (apaQTLs) and ambiguous (Fig 3B). Our aim was to distinguish primary splicing and

transcript-level QTLs, where the putative causal variant has a direct effect on promoter or splice junction choice, from likely secondary associations, where the change in promoter, transcript or splice junction usage is likely a consequence of large-effect eQTLs, perhaps by altering the kinetics or fidelity of splicing [33,34]. Although this process is inherently subjective and noisy, we have provided some examples in S2 Text to clarify the implicit decision rules that we used. As an example, we detected a splicing QTL affecting the length of exon 4 of *CYP2R1* (Fig 2). *CYP2R1* is highly likely to be the causal gene at this locus as it codes for the Cytochrome P450 2R1 microsomal vitamin D 25-hydroxylase [35]. We found that although transcript-level methods (tx, txrevise and leafcutter) detected at least one colocalisation for 37/53 independent signals, only 14 of those (7 puQTLs, 6 sQTLs and 1 apaQTL) could be classified as primary transcript-level QTLs (Fig 3B). Other 23 cases were either ambiguous or could be better explained by strong primary eQTL effects that led to small downstream changes in splicing or transcript usage (S3 Table, Fig C in S1 Text).

Even though Leafcutter detected all seven visually confirmed sQTLs and 5/7 puQTLs, it also detected 11 additional signals, nine of which could be better explained by a strong eQTL effects (e.g. *CELSR2* eQTL at the *SORT1* locus (Fig D in S1 Text)). Thus, the fact that colocalisation is detected by one of the transcript-level methods (tx, txrevise or leafcutter) does not reliably indicate that the underlying signal is driven by a primary splicing mechanism. The visualisations also helped us to detect three likely cases of reference mapping bias at the *DHCR7*, *NUDT9* and *JUND* genes (Figs E-G in S1 Text). For discussion of why we opted not to correct for reference mapping bias during molecular trait quantification, see S1 Text.

We also noticed that 15/32 confirmed eQTLs colocalised with more than one gene (Fig 3C). In contrast, only one of seven puQTLs and one of six sQTLs colocalised with multiple genes, suggesting that sQTLs and puQTLs might be less pleiotropic than eQTLs. To evaluate if lower pleiotropy also translated into more accurate causal gene prioritisation, we manually reviewed all of the 53 GWAS signals to identify the most likely causal genes. We integrated information about missense variant associations, gene presence in the vitamin D synthesis pathway and other literature evidence to prioritise the most likely causal gene for 28/53 GWAS signals (S3 Table). For four of the six sQTL signals, the colocalising gene overlapped the prioritised causal gene (*CYP2R1*, *HAL*, *GC* and *SDR42E1)* and for two signals we could not prioritise the causal gene. For eQTLs, we prioritise the most likely causal gene at 19/32 loci. In 11/19 cases (3 shared with sQTLs, Fig 3B) the colocalising eQTL genes completely overlapped the prioritised genes. In four cases the prioritised gene (*SORT1*, *FLG*, *HAL*, *CETP*) was one of multiple co-localizing genes. Finally, at four additional signals, the prioritised gene was different from the one that had eQTL colocalisation evidence (S3 Table). Interestingly, in three of the four cases the GWAS lead variant was a missense (*SEC23A*, *PLA2G3*) or a synonymous variant (*CYP2R1)* in the prioritised gene. While the number of loci observed here is small, these results suggest that while visually confirmed sQTLs colocalise with a smaller fraction of GWAS loci than eQTLs (6 vs 32), they are also less pleiotropic and thus more likely to identify the correct causal gene.

## Discussion

We have made three major changes to the eQTL Catalogue in release 6. First, we have integrated data from ten additional eQTL studies bringing the total number of unique eQTL datasets to 127. These datasets contain uniformly processed results from 30,602 samples from 7,526 individuals. We have also updated our genotype imputation, RNA-seq analysis and QTL mapping workflows to add support for the X chromosome, added Leafcutter as a splicing quantification method and added support for fine mapping-based colocalisation analysis with

coloc.susie [13]. Finally, we have developed static QTL coverage plots to visualise molecular QTL associations at the level of RNA-seq read alignments. All of our results and data are available on the eQTL Catalogue FTP server and REST API.

To quantify the impact of these updates, we performed colocalisation between all molecular QTLs present in the eQTL Catalogue and fine mapped GWAS signals for plasma vitamin D levels in the UK Biobank [36]. The QTL coverage plots allowed us to assign an empirical functional consequence (eQTL, sQTL, puQTL, apaQTL) for 42/53 colocalising loci while 11 remained ambiguous. This revealed that while primary sQTLs explained fewer GWAS signals than eQTLs, they also appeared to be less pleiotropic and more likely to identify the correct target genes. A limitation of our approach is that we used manual visual inspection to assign mechanisms to different types of molecular QTLs. Although we tried to be careful, there is a small risk that this approach could have introduced inadvertent confirmation bias (e.g. classifying less pleiotropic loci as sQTLs). We expect that it might be possible to automate this classification in the future by machine learning approaches that take into account variant-level annotations such as splicing scores [37,38] or distance to genomic features.

We also observed that while most GWAS signals colocalised with an eQTL, approximately ~50% of the eQTL colocalisations prioritised more than one gene. Similarly in 4/19 cases, the colocalising gene was different from the manually prioritised causal gene. This agrees with multiple previous observations that eQTL colocalisation alone often achieves low precision in causal gene identification [39,40]. This does not seem to be a simple artefact of colocalisation analysis as CRISPR experiments have also revealed that targeting a single enhancer often regulates the expression of multiple target genes [41–43]. We believe that while eQTL colocalisation can sometimes reveal trait-relevant tissues or cell types, target gene identification requires integration of multiple strands of evidence. Taking into account variants with potentially less pleiotropic effects such as missense and splice variants can also be helpful.

The systematic re-analysis and visualisation of molecular QTLs presented here would not have been possible without the researchers of the 31 original studies making their individual-level gene expression and genotype data available for qualified researchers. We are committed to sharing all summary statistics and fine mapping results openly and will seek to continuously integrate new eQTL datasets as they become available. We are also working on making the static QTL coverage plots available via an API and an interactive web interface.

## Methods

### Ethics statement

For all newly added datasets, we applied for access via the relevant Data Access Committees. The database accessions and contact details of the individual Data Access Committees can be found on the eQTL Catalogue website (http://www.ebi.ac.uk/eqtl/Studies/). In our applications, we explained the project and our intent to share the association summary statistics publicly. Ethical approval for the project was obtained from the Research Ethics Committee of the University of Tartu (approval 287/T-14).

### Genotype data

**Pre-imputation quality control.** We lifted coordinates of the genotyped variants to the GRCh38 build with CrossMap v0.4.1 [44]. We aligned the strands of the genotyped variants to the 1000 Genomes 30x on GRCh38 reference panel [26] using Genotype Harmonizer [45]. We excluded genetic variants with Hardy-Weinberg p-value $< 10^{-6}$, missingness $> 0.05$ and minor allele frequency $< 0.01$ from further analysis. On the X chromosome, we applied the

QC filters to female samples only and then retained the same variants also in the male samples. We also excluded samples with more than 5% of their genotypes missing.

**Genotype imputation and quality control.** We pre-phased and imputed the microarray genotypes to the 1000 Genomes 30x on GRCh38 reference panel [26] using Eagle v2.4.1 [46] and Minimac4 [47]. On the X chromosome, we performed imputation separately for variants located in the pseudoautosomal (PAR) and non-PAR regions. After imputation, we multiplied male genotype dosage in the non-PAR region by two to ensure that it is on the same scale with the female genotypes. We used bcftools v1.9.0 to exclude variants with minor allele frequency (MAF) < 0.01 and imputation quality score R2 < 0.4 from downstream analysis. The genotype imputation and quality control steps are implemented in eQTL-Catalogue/genimpute (v22.01.1) workflow available from GitHub.

We aligned the low-coverage whole genome sequencing (WGS) data from the BLUEPRINT project to the GRCh38 reference genome with bwa v0.7.17 [48] and performed imputation to the 1000 Genomes 30x on GRCh38 reference panel using GLIMPSE v1.1.1 [49]. The low-coverage WGS genotype imputation workflow is available from GitHub: https://github.com/peepkolberg/glimpse.

## Phenotype data

**Studies.** eQTL Catalogue release 6 contains phenotype data from the following 25 RNA-seq studies: ROSMAP [50], BrainSeq [51], TwinsUK [52], FUSION [53], BLUEPRINT [54,55], Quach_2016 [56], Schmiedel_2018 [57], GENCORD [58], GEUVADIS [59], Alasoo_2018 [11], Nedelec_2016 [60], Lepik_2017 [61], HipSci [62], van_de_Bunt_2015 [63], Schwartzentruber_2018 [64], GTEx v8 [3], CAP [19], Peng_2018 [24], PhLiPS [22], iPSCORE [65], CommonMind [16], Braineac2 [15], Steinberg_2020 [25], Young_2019 [23], Bossini-Castillo_2019 [18]. It also contains data from the following 7 microarray studies: CEDAR [66], Fairfax_2012 [67], Fairfax_2014 [68], Kasela_2017 [69], Naranbhai_2015 [70], Kim-Hellmuth_2017 [20] and Gilchrist_2021 [17].

**Quantification.** We quantified transcription at five different levels: (1) gene expression, (2) exon expression, (3) transcript usage, (4) transcriptional event usage, and (5) splice-junction usage (Fig A in S1 Text). Quantification was performed using version v22.05.1 of the eQTL-Catalogue/rnaseq workflow implemented in Nextflow [71]. Before quantification, we used Trim Galore v0.5.0 to remove sequencing adapters from the fastq files.

For gene expression quantification, we used HISAT2 v2.2.1 [72] to align reads to the GRCh38 reference genome (Homo_sapiens.GRCh38.dna.primary_assembly.fa file downloaded from Ensembl). We counted the number of reads overlapping the genes in the GENCODE V39 [73] reference transcriptome annotations with featureCounts v1.6.4 [74]. To quantify exon expression, we first created an exon annotation file (GFF) using GENCODE V39 reference transcriptome annotations and `dexseq_prepare_annotation.py` script from the DEXSeq [75] package. We then used the aligned RNA-seq BAM files from the gene expression quantification and featureCounts with flags `'-p -t exonic_part -s ${direction} -f -O'` to count the number of reads overlapping each exon.

We quantified transcript and event expression with Salmon v1.8.0 [76]. For transcript quantification, we used the GENCODE V39 (GRCh38.p13) reference transcript sequences (fasta) file to build the Salmon index. For transcriptional event usage, we downloaded pre-computed txrevise [5,28] alternative promoter, splicing and alternative 3′ end annotations corresponding to Ensembl version 105 from Zenodo (https://doi.org/10.5281/zenodo.6499127) in GFF format. These annotations had been augmented with additional experimentally derived promoter annotations from the FANTOM5 consortium [77,78]. We then used gffread [79] to

generate fasta sequences from the event annotations and built Salmon indices for each event set as we did for transcript usage. Finally, we quantified transcript and event expression using `salmon quant` with '`-seqBias -useVBOpt -gcBias -libType`' flags. All expression matrices were merged using csvtk v0.17.0. Our reference transcriptome annotations are available from Zenodo (https://doi.org/10.5281/zenodo.4715946).

For Leafcutter analysis, splice junctions of the aligned reads were extracted using the *junctions extract* command of the regtools v0.5.2 [80] with options '`-s $strand -a 8 -m 50 -M 500000`'. Then, these splice-junctions were clustered using leafcutter_cluster_regtools. py script from LeafCutter v0.2.9 with options '`-m 50 -o leafcutter -l 500000 -checkchrom=True`'.

**Normalisation.** We normalised the gene and exon-level read counts using the conditional quantile normalisation (cqn) R package v1.30.0 [81] with gene or exon GC nucleotide content as a covariate. We downloaded the gene GC content estimates from Ensembl biomaRt and calculated the exon-level GC content using bedtools v2.19.0 [82]. We also excluded lowly expressed genes, where 95 per cent of the samples within a dataset had transcripts per million (TPM)-normalised expression less than 1. To calculate transcript and transcriptional event usage values, we obtained the TPM normalised transcript (event) expression estimates from Salmon. We then divided those transcript (event) expression estimates by the total expression of all transcripts (events) from the same gene (event group). Subsequently, we used the inverse normal transformation to standardise all five molecular quantification estimates. Normalisation scripts together with containerised software are publicly available at https://github.com/eQTL-Catalogue/qcnorm.

## Association testing and statistical fine mapping

We performed association testing separately in each dataset and used a +/- 1 megabase *cis* window centred around the start of each gene. First, we excluded molecular traits with less than five genetic variants in their *cis* window, as these were likely to reside in regions with low genotyping coverage. We also excluded molecular traits with zero variance across all samples and calculated phenotype principal components using the prcomp R stats package (center = true, scale = true). We calculated genotype principal components using plink2 v1.90b3.35. We used the first six genotype and molecular trait principal components as covariates in QTL mapping. We calculated nominal eQTL summary statistics using the GTEx v6p version of the FastQTL [83] software (https://github.com/francois-a/fastqtl) that also estimates standard errors of the effect sizes. We used the '`-window 1000000 -nominal 1`' flags to find all associations in 1 Mb *cis* window. For permutation analysis, we used QTLtools v1.3.1 [84] with '`-window 1000000 -permute 1000 -grp-best`' flags to calculate empirical p-values based on 1000 permutations. The '`-grp-best`' option ensured that the permutations were performed across all molecular traits within the same 'group' (e.g. multiple probes per gene in microarray data or multiple transcripts or exons per gene in the exon-level and transcript-level analysis) and the empirical p-value was calculated at the group level.

We performed QTL fine mapping using the Sum of Single Effects Model (SuSiE) [14] implemented in the susieR v0.11.92 R package. We converted the genotypes from VCF format to a tabix-indexed dosage matrix with bcftools v1.10.2. We imported the genotype dosage matrix into R using the Rsamtools v2.8.0 R package. We used the same normalised molecular trait matrix used for QTL mapping. We regressed out the first six phenotype and genotype PCs separately from the phenotype and genotype matrices. We performed fine mapping with the following parameters: L = 10, estimate_residual_variance = TRUE, estimate_prior_variance = TRUE, scaled_prior_variance = 0.1, compute_univariate_zscore = TRUE, min_abs_corr = 0. The steps

described above are implemented in the eQTL-Catalogue/qtlmap v22.04.01 Nextflow workflow available from GitHub.

## Filtering of transcript-level summary statistics

We filtered transcript-level summary statistics using a connected components approach [85] to select the strongest signals per transcript-level group (gene for transcript and exon level, clusters for leafcutter). For each group, first, we filtered out the credible sets where maximum absolute z value is lower than 3 and size is bigger than 200 variants. Then, we found overlapping variants between credible sets, defining these credible sets as connected components. For each connected component we selected the molecular trait with the highest posterior inclusion probability (PIP) as the 'tag' trait and kept only the summary statistics of these selected molecular traits. We provided three specific examples for genes *SDR42E1*, *HAL* and *CYP2R1* on Figs H-M in S1 Text to illustrate how keeping one 'tag' molecular trait for each gene can faithfully capture the transcript-level QTL signals detect at these loci while significantly reducing the size of the summary statistics files.

## Colocalisation with vitamin D GWAS

We used coloc.susie [13] to perform signal-level colocalisation between all RNA-seq-based datasets in the eQTL Catalogue and GWAS summary statistics for vitamin D levels in the UK Biobank. For all molecular QTLs, we used the log Bayes factors (LBFs) exported by our eQTL-Catalogue/qtlmap v22.04.01 workflow. For the vitamin D GWAS, we used published SuSiE fine mapping results from a previous study [36] downloaded from Google Cloud (link). We performed colocalisation between all pairs of independent fine mapped signals (up to 10 per locus) and reported results where PP4 > 0.9. The colocalisation workflows is available from GitHub (https://github.com/ralf-tambets/coloc).

## Generation of QTL coverage plots

We used the bamCoverage command from deepTools v3.2.0 [86] with bin-size option '`-bs 5`' to generate read-coverage (bigwig) files. We then used extractCoverageData and plotCoverageData commands of wiggleplotr R v1.13.1 package [29] to read specific regions of the bigwig files, scale all introns to the length of 50 nucleotides, and generate the plots as ggplot2 [87] objects. Finally, we generated exon QTL effect-size plots with ggplot2 v3.3.6 and put all the plots together with the cowplot v1.1.1 R package [88]. We used *tabix.read.table* from seqminer v8.4 package [89] to extract both genotype and QTL data from indexed files in the regions of interest. Coverage plot generation workflow is publicly available at https://github.com/kerimoff/leafcutter_plot.

## Supporting information

**S1 Text. Supplementary Notes and Figures.**
(PDF)

**S2 Text. Classification criteria for QTL coverage plots.**
(PDF)

**S1 Table. Metadata for datasets included in the eQTL Catalogue release 6.**
(XLSX)

**S2 Table. Availability of X chromosome genotypes.**
(XLSX)

**S3 Table. Classification of vitamin D colocalisation results.**
(XLSX)

# Acknowledgments

The RNA-seq quantification and QTL analyses were performed at the High Performance Computing Center, University of Tartu. We thank O.E. Oopkaup, S. Kuusemets and the rest of the team of the High Performance Computing Center for their professional and timely technical support in enabling the analyses performed in this paper. We thank M. Weale for preparing the Braineac2 dataset for analysis.

# Author Contributions

**Conceptualization:** Nurlan Kerimov, James D. Hayhurst, Helen Parkinson, Kaur Alasoo.

**Data curation:** Nurlan Kerimov, James D. Hayhurst, Peep Kolberg, Anshika Chowdhary, Andreas Vija, Hans J. Teras, Masahiro Kanai, Jacob Ulirsch, Mina Ryten, John Hardy, Sebastian Guelfi, Daniah Trabzuni, Sarah Kim-Hellmuth, Hilary Finucane, Kaur Alasoo.

**Formal analysis:** Nurlan Kerimov, Ralf Tambets, Ida Rahu, Peep Kolberg, Anshika Chowdhary, Andreas Vija, Masahiro Kanai, Jacob Ulirsch.

**Funding acquisition:** William Rayner, Hedi Peterson, Helen Parkinson, Kaur Alasoo.

**Investigation:** Nurlan Kerimov.

**Methodology:** Nurlan Kerimov, Ralf Tambets, James D. Hayhurst, Ida Rahu, Peep Kolberg, Kaur Alasoo.

**Project administration:** Hedi Peterson, Abayomi Mosaku, Helen Parkinson, Kaur Alasoo.

**Resources:** James D. Hayhurst, Masahiro Kanai, Jacob Ulirsch, Mina Ryten, John Hardy, Sebastian Guelfi, Daniah Trabzuni, Sarah Kim-Hellmuth, William Rayner, Hilary Finucane.

**Software:** Nurlan Kerimov, Ralf Tambets, James D. Hayhurst, Peep Kolberg, Uku Raudvere, Ivan Kuzmin, Andreas Vija.

**Supervision:** William Rayner, Hilary Finucane, Hedi Peterson, Abayomi Mosaku, Helen Parkinson, Kaur Alasoo.

**Validation:** Ida Rahu.

**Visualization:** Nurlan Kerimov.

**Writing – original draft:** Nurlan Kerimov, Kaur Alasoo.

**Writing – review & editing:** Nurlan Kerimov, Kaur Alasoo.

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
