## [Decision Letter · Decision Letter 0]

13 Jun 2023

Dear Dr Alasoo,

Thank you very much for submitting your Research Article entitled 'Systematic visualisation of molecular QTLs reveals variant mechanisms at GWAS loci' to PLOS Genetics.

The manuscript was fully evaluated at the editorial level and by independent peer reviewers. The reviewers appreciated the attention to an important topic but identified some concerns that we ask you address in a revised manuscript.

We therefore ask you to modify the manuscript according to the review recommendations. Your revisions should address the specific points made by each reviewer.

Yours sincerely,

Zoltán Kutalik, PhD

Academic Editor

PLOS Genetics

Hua Tang

Section Editor

PLOS Genetics

As the authors can see, the reviewers were generally positive of the manuscript, but raised some important points to improve it.

Beyond those comments, I’d like to emphasise that the title and the abstract of the paper should be much better defined. The title could be something more along the line of “Updates from the eQTL Catalogue resource” and the abstract should reflect the message of the paper, i.e. talking about the novel aspect of the resource, covering the major chapters of the results, including the case study. Thus, I suggest that the authors completely rewrite the abstract and the title.

Reviewer's Responses to Questions

**Comments to the Authors:**

Reviewer #1: The authors present an update to the eQTL Catalogue, including new studies, inclusion of X chromosome eQTLs, as well as a new approach to discovery of expression/splicing QTLs that helps to identify most relevant signals within a gene, and de-prioritize other, correlated signals which may be less likely to be driving downstream phenotypes of interest. The resource, methods, and discussion are valuable for researchers in the fields of QTL discovery, and for QTL-GWAS colocalization efforts, with an emphasis on quality assessment via visual inspection of expression/splicing information within loci.

Comments:

1) The authors mention that the various transcript-level association statistics are correlated and so have a large amount of redundancy that can be addressed with fine-mapping-based filtering. For a subset of genes, could the authors provide example correlation matrices, just for the reader to have an extent of the correlation of features being considered.

2) Also related to the fine-mapping-based filtering: "only retain the most strongly associated molecular trait", is described in the Methods as using the highest PIP among a connected component formed over credible set input. If some features tend to have higher PIP values (across the whole dataset) does this shift the filtering toward those features? Is this related to the underling p-value distribution for the features? This comment is not suggesting to change the underlying methods or analyses. I am just curious do the authors think that considering the ranks within feature, rather than the raw PIP, would provide a different filtering outcome (that label of the molecular trait would be different)? This matters mostly, it would appear, for downstream tallying of mechanisms.

3) Is the coverage in Figure 2A log scale and is this why the genotypes 0 and 1 are closer to each other than 1 and 2?

4) While the authors note that manual inspection was used to assign mechanisms to different types of molecular QTL, it would be helpful to visualize the implied rules. Could the authors provide a table or a Sankey diagram linking the molecular QTL combinations to the empirical functional consequences?

5) Regarding the multiple target genes in 3C, are these ever resolvable by examining the number of signals per gene? E.g. if a gene has two colocalized signals that may be a better candidate than another gene with only one colocalized signal.

Reviewer #2: This study by Kerimov et al provides an update to the eQTL Catalogue to facilitate downstream characterization of the exact mechanism of action at each molecular QTL (regulating expression, splicing, promoter usage, alternative polyadenylation, etc). This task of characterization has been a challenging problem for a number of reasons (for example, the substantial overlap in QTLs detected for gene expression, exon expression, txrevise, leafcutter, and transcript usage). This study presents an approach to generate QTL coverage plots for genetic signals of molecular traits to overcome these challenges.

Although this study is potentially useful to the research community (such as in supporting the development of new colocalization methods that assume multiple independent causal genetic effects), I have some major concerns about the way the study presents its methodology and results.

1. The Title, Introduction and Results all appear to show the relevance of the approach for a generic molecular QTL. However, the abstract itself restricts itself to splicing. The focus of the paper is thus quite confusing.

2. To add to the confusion, the first results presented concern the number of eGenes for the X chromosome. While I commend the authors for including these results on a generally neglected part of the genome, there needs to be more (and better) contextualization of these findings. The primary aim of the study, after all, is the development of a "systematic visualisation of molecular QTLs" for the purpose of enhancing the identification of the variant mechanisms at GWAS loci.

3. Figure 1c shows that the newly added datasets since the previous release appear to be low sample size datasets. Therefore, rather than showing the number of eGenes from the previous release and from the new datasets, it would be important to show how many new eGenes were actually identified through inclusion of these new (low sample size) datasets. What was gained from their inclusion?

4. Finally, without a precise description of the statistical methodology for classifying a signal (into an expression QTL, promoter usage QTL, splicing QTL, etc.) -- given the potentially substantial sharing of significantly colocalized signals, as in the example of vitamin D (figure 3) -- the value/quality of the "visualization" approach and the resulting resource of QTL coverage plots is really quite challenging to assess. I worry that without a rigorous statistical framework to guide interpretation, the resource will have very limited impact.

**Have all data underlying the figures and results presented in the manuscript been provided?**

Reviewer #1: Yes

Reviewer #2: Yes

PLOS authors have the option to publish the peer review history of their article (what does this mean?). If published, this will include your full peer review and any attached files.

Reviewer #1: **Yes: **Michael Love

Reviewer #2: No

---

## [Decision Letter · Decision Letter 1]

22 Aug 2023

Dear Dr Alasoo,

We are pleased to inform you that your manuscript entitled "eQTL Catalogue 2023: new datasets, X chromosome QTLs, and improved detection and visualisation of transcript-level QTLs" has been editorially accepted for publication in PLOS Genetics. Congratulations!

Yours sincerely,

Zoltán Kutalik, PhD

Academic Editor

PLOS Genetics

Hua Tang

Section Editor

PLOS Genetics

Comments from the reviewers (if applicable):

Reviewer's Responses to Questions

**Comments to the Authors:**

Reviewer #1: The authors have entirely addressed my previous comments. The manuscript should be a helpful guide for interpretation of transcript- and splice-level QTLs. In particular the detailed examples in S2 are a great addition.

Reviewer #2: The authors have substantially improved the clarity of the manuscript through the edits that have been made. The new title and revised abstract more accurately reflect the focus of the study and help to clarify its aims. Although a precise description of the statistical methodology for assigning mechanisms to molecular QTLs is not fully worked out (one of my original concerns), the examples that the authors provide in Supplementary Text 2 are informative and do present the implicit "decision rules". Overall, I believe the new datasets and the visualization plots in the updated eQTL Catalogue will be very useful to the community.

**Have all data underlying the figures and results presented in the manuscript been provided?**

Reviewer #1: Yes

Reviewer #2: Yes

PLOS authors have the option to publish the peer review history of their article (what does this mean?). If published, this will include your full peer review and any attached files.

Reviewer #1: No

Reviewer #2: **Yes: **Eric R. Gamazon

**Data Deposition**

http://datadryad.org/submit?journalID=pgenetics&manu=PGENETICS-D-23-00444R1

**Press Queries**

---

## [Editor Report · Acceptance letter]

14 Sep 2023

PGENETICS-D-23-00444R1 

eQTL Catalogue 2023: new datasets, X chromosome QTLs, and improved detection and visualisation of transcript-level QTLs 

Dear Dr Alasoo, 

We are pleased to inform you that your manuscript entitled "eQTL Catalogue 2023: new datasets, X chromosome QTLs, and improved detection and visualisation of transcript-level QTLs" has been formally accepted for publication in PLOS Genetics! Your manuscript is now with our production department and you will be notified of the publication date in due course.

With kind regards,

Judit Kozma

PLOS Genetics

On behalf of:
